# Comorbidity Risk Score in Association with Cancer Incidence: Results from a Cancer Screenee Cohort

**DOI:** 10.3390/cancers12071834

**Published:** 2020-07-08

**Authors:** Tung Hoang, Jeonghee Lee, Jeongseon Kim

**Affiliations:** Department of Cancer Biomedical Science, National Cancer Center Graduate School of Cancer Science and Policy, Goyang 10408, Korea; 75256@ncc.re.kr (T.H.); jeonghee@ncc.re.kr (J.L.)

**Keywords:** comorbidity risk score, dietary score, cancer incidence, Gaussian graphical model

## Abstract

The combined effects of comorbidities can cause cancer incidence, while the effects of individual conditions, alone, might not. This study was conducted to investigate the joint impact of comorbidities on cancer incidence. The dietary score for energy-adjusted intake was calculated by applying a Gaussian graphical model and was then categorized into tertiles representing light, normal, and heavy eating behaviors. The risk point for cancer, according to the statuses of blood pressure, total cholesterol, fasting glucose, and glomerular filtration rate was computed from a Cox proportional hazard model adjusted for demographics and eating behavior. The comorbidity risk score was defined as the sum of the risk points for four comorbidity markers. We finally quantified the hazard ratios (HRs) and 95% confidence intervals (CIs) for the association between the strata of the comorbidity risk score and cancer incidence. A total of 13,644 subjects were recruited from the Cancer Screenee Cohort from 2007–2014. The comorbidity risk score was associated with cancer incidence in a dose-dependent manner (HR = 2.15, 95% CI = 1.39, 3.31 for those scoring 16–30 vs. those scoring 0–8, *P*-trend < 0.001). Subgroup analysis still showed significant dose-dependent relationships (HR = 2.39, 95% CI = 1.18, 4.84 for males and HR = 1.99, 95% CI = 1.11, 3.59 for females, *P*-trend < 0.05). In summary, there was a dose-dependent impact of comorbidities on cancer incidence; Highlights: Previous studies have generally reported that hypertension, hypercholesterolemia, diabetes, and chronic kidney disease might predispose patients to cancer. Combining these chronic diseases into a single score, this study found a dose-dependent association between the data-driven comorbidity risk score and cancer incidence.

## 1. Introduction

Cancer has been a global health issue over the past few decades. According to GLOBOCAN, there were an estimated 18.1 million new cancer cases and 9.6 million cancer-related deaths in 2018 [1]. In addition to cancer, other noncommunicable diseases, including cardiovascular disease (CVD), chronic respiratory diseases, and diabetes, are leading causes of death worldwide [2,3]. Although cancer and these chronic conditions are known to share common risk factors, certain comorbidities might independently contribute to the development of cancer [4]. To date, several studies have found a large number of risk factors for cancer. The International Agency for Research on Cancer, which classified these carcinogens into different groups based on their relationship with cancer risk, did not consider comorbidity to be a risk factor for cancer [5]. Additionally, the cancer prevention program mostly focuses on lifestyle risk factors such as nutrition and physical activity [4].

Previous studies have generally investigated the association between separate chronic diseases or disease markers and the risk of cancer [6,7,8,9,10,11,12,13,14,15]. For example, hypertension was found to be associated with an increased risk of kidney cancer [6,7,8], possibly due to the downregulation of the angiotensin-converting enzyme [9]. Several meta-analyses have consistently reported a higher cancer risk among those with diabetes mellitus [10,11,12]. Furthermore, there is also an elevated risk of cancer among those with high cholesterol levels and chronic kidney disease [13,14,15].

However, comorbidities may jointly impact the risk of cancer, and research on this topic remains limited. Tu et al. recently reported the chronic disease risk score, which summarizes eight different chronic diseases and disease markers to elucidate the contribution of comorbidities to the risk of cancer [4]. In that large population-based prospective study of 405,878 Taiwanese participants, subjects with the highest comorbidity score were associated with 121% of the risk of incident cancer (hazard ratio (HR) = 2.21, 95% confidence interval (CI) = 1.77, 2.75) [4]. Although dietary behavior was found to play an important role in the development of several chronic diseases and cancer [16,17,18,19,20], only the assessment of fruit and vegetable intake was included in the study by Tu et al. [4] In this study, we assessed dietary behavior through whole diet consumption instead of a single food item approach. The partial correlation among food groups was accounted for in the data-driven dietary network.

Given that lifestyle factors and disease structure differ by country, and that dietary behavior may affect cancer incidence, this study aims to evaluate the associations between the joint influence of comorbidities and cancer incidence among the Korean population.

## 2. Materials and Methods

### 2.1. Study Population

The details of the Cancer Screenee Cohort are described elsewhere [21]. Between October 2007 and July 2014, a total of 13,644 subjects completed the written consent form and volunteered to provide information on their medical history, clinical test results, and dietary consumption (Figure 1). Those with missing data on dietary intake or with unrealistic data for energy intake (<500 or >4000 kcal) were excluded. Among the 8597 individuals identified at baseline, 612 subjects who were previously diagnosed with cancer and 2342 subjects who had missing values regarding demographics and comorbidities were additionally excluded. Because cancer has a long latency, it was possible for a tumor to be present but clinically undetected at the time of participation in the study. Hence, we excluded subjects diagnosed with cancer within 1 year to minimize reverse causality. As a result, a total of 5606 subjects were included in the final analysis. The Institutional Review Board of the National Cancer Center approved this study protocol (number NCCNCS-07-077).

### 2.2. Variable definition

Cancer incidence was identified via diagnosis and classified according to the International Classification of Diseases-10 codes. Time to cancer incidence was defined as the interval between the date of study enrollment and the date of receiving a new diagnosis of cancer. The censoring date was defined as the last follow-up date (31 December 2016) or the date of death from a noncancer cause.

Regarding comorbidities, we selected blood pressure, total cholesterol, fasting glucose, and the glomerular filtration rate (GFR), together with self-reported hypertension and diabetes. The Joint National Committee guidelines recommended dividing blood pressure (mmHg) into normal (<120/80, not receiving therapy and not previously diagnosed with hypertension), prehypertension (120–139/80–89, not receiving therapy and not previously diagnosed with hypertension), and hypertension (≥140/90, receiving therapy or previously diagnosed with hypertension) [22]. Total cholesterol (mg/dL) was divided into low (<180, not taking drugs and not previously diagnosed with dyslipidemia), normal (180–200, not taking drugs and not previously diagnosed with dyslipidemia), and elevated (>200, taking drugs or previously diagnosed with dyslipidemia) [23]. Fasting glucose (mg/dL) was classified as normal (<110, no underlying treatment and not previously diagnosed with diabetes) or prediabetes and diabetes (≥110, underlying treatment or previously diagnosed with diabetes). The GFR, which was calculated by using the Modification of Diet in Renal Disease equation [24], was categorized into ≥90, 60–89, and <60 mL/min/1.73 m^2^.

The covariates included age (years), sex (male and female), marital status (married, cohabitant and others), education (<high school, high school graduate, and ≥college), monthly income (<2 million, 2–4 million, and ≥4 million South Korean won (KRW)), smoking (never, past, and current), drinking (never, past, and current), physical activity (yes and no), body mass index (BMI) (<23, 23–24.9, and ≥25 kg/m^2^), and eating behavior (light, normal, and heavy). Details of these co-factors of medical history are summarized in Table A1 and Table A2.

### 2.3. Statistical Analysis

Figure A1 shows how the dietary intake score was calculated. Briefly, we applied a Gaussian graphical model (GGM) to identify the dietary network, which was composed of 16 food groups as nodes and their pairwise partial correlations as edges. The network structure was estimated using the extended Bayesian information criteria (EBIC) model selection set at 0.5 [25,26]. The eigenvector centrality of the GGM-identified network was computed as the weight of the nodes [27], and the dietary intake score was defined as the sum of the amount consumed in each food group with their respective weights. The dietary intake score therefore indicates the total energy-adjusted intake—in grams/day—for the whole diet of the 16 food groups; therefore, a higher dietary intake score represented a higher amount of dietary consumption (g/day) after adjusting for the weights of the 16 food groups. The dietary intake score was then categorized into low, medium, and high tertiles, which correspond to light, normal, and heavy eating behaviors, respectively.

We computed the comorbidity risk score following the procedure in Sullivan et al. [28], which was developed based on the Heart Framingham Study. Briefly, we used a Cox regression model to obtain the coefficient of cancer incidence related to each chronic condition, adjusting for age and other demographic variables. Then, the risk point corresponding to each chronic condition was obtained by dividing the above-calculated regression coefficients for comorbidities by the coefficient for a one-year increase in age. The comorbidity score for each individual was defined as the sum of the four risk points for four chronic conditions. The differences between the cancer and noncancer groups and among groups with different comorbidity scores according to anthropometric factors and the 16 food groups were explored using chi-square tests for categorical variables and t-tests and ANOVA for continuous variables. The determination of the time to cancer incidence according to the comorbidity risk score was made via Kaplan–Meier estimates. The association between the comorbidity risk score and cancer incidence was finally investigated using the Cox proportional hazards model. We assigned the mean values of the comorbidity risk score to test the linear trend across the strata.

Given that a previous study did not support the association between total cholesterol and cancer risk [29], we excluded total cholesterol from the comorbidity risk score in the sensitivity analysis. We also performed the analysis while considering all the participants, regardless of the time to cancer diagnosis, and compared the results with our main findings.

## 3. Results

### 3.1. Dietary Scores of Study Participants

The partial correlation network of dietary intake was identified from the GGM (Figure 2). The partial correlation of diet consumption (g/day) between the two food groups controlling for remaining food groups in the network is reported in Table 1. The strongest regularized partial correlation was observed between ‘oils and fats’ and ‘sugars and sweets’ (0.70), followed by ‘seasonings’ and ‘potatoes and starches’ (0.37) or ‘vegetables’ (0.34).

The amount of daily consumption and the weights of the 16 food groups are reported in Table 2. The intake of 16 food groups was not significantly different between cancer incidence and noncancer groups (*p* > 0.05). The total energy-adjusted intake was 295.1 ± 69.0, 483.2 ± 54.5, and 834.4 ± 276.1 (g/day), for subjects with light, normal, and heavy eating behaviors, respectively.

Nodes reflect food groups, and edges reflect the conditional dependencies between food groups. Green lines show positive partial correlations, and red lines show negative partial correlations. The thickness of the edges represents the strength of correlations.

### 3.2. Comorbidity risk scores of study participants

Table 3 presents the associations between individual comorbidities and cancer incidence. A borderline increased incidence of cancer was observed in those with hypertension (adjusted HR = 1.53, 95% CI = 1.02, 2.29). The significant associations in the model fully adjusted for all possible covariates were observed for hypertension (HR = 1.56, 95% CI = 1.02, 2.39) and normal total cholesterol (HR = 1.51, 95% CI = 1.03, 2.19). The comorbidity risk scores were highest in subjects with a GFR ≥90 mL/min/1.73 m^2^ (score = 9). Intermediate risk scores were assigned to those with hypertension (score = 8) and normal cholesterol (score = 8), followed by low cholesterol (score = 5) and prediabetes and diabetes (score = 5). Subjects with elevated blood pressure and GFR 60–89 mL/min/1.73 m^2^ were at low risk (score = 2).

### 3.3. Baseline Characteristics of the Study Participants

During the median follow-up time of 5.34 years (interquartile range (IQR) = 4.03–6.45, total 29,145 person-years), 176 patients were newly diagnosed with cancer. Cancer subjects were observed to be significantly older than noncancer participants, with age at baseline in the two groups of 55.8 ± 8.7 years and 52.5 ± 8.2, respectively (*p* < 0.001). Except employment status (*p* = 0.02), other demographic characteristics and GGM-identified dietary scores were not significantly different between the cancer and noncancer groups (Table 4). In contrast, most of the factors, including age, sex, education, employment, tobacco smoking, alcohol consumption, and BMI, were unequally distributed among groups stratified by comorbidity risk scores (*p* < 0.001).

### 3.4. Comorbidity and Cancer Incidence

Figure 3 shows the Kaplan–Meier estimates for the cancer-free probability for different strata of comorbidity risk scores. The median times to cancer incidence were 5.42 years (IQR = 4.24, 6.53), 5.44 years (IQR = 4.12, 6.45), 5.34 years (IQR = 3.87, 6.44), and 4.38 years (IQR = 3.23, 6.15) for scores of 0–8, 9–10, 11–15, and 16–30, respectively.

The association between the comorbidity risk score and cancer incidence is detailed in Table 5. Compared with participants scoring 0–8, those whose scores were 16–30 had a 127% increased risk (HR = 2.27, 95% CI = 1.49, 3.46) of cancer in the crude model. A significantly positive association was still observed in the fully adjusted model, with HR = 2.15, 95% CI = 1.39, 3.31. The linear trend test suggested that there might be a dose-dependent statistical association between the comorbidity risk score and cancer incidence (*p* < 0.001).

In the subgroup analysis by sex, the cancer incidence was higher among subjects who scored 16–30 than among those who scored 0–8, with HRs (95% CIs) of 2.39 (1.18, 4.84) for males and 1.99 (1.11, 3.59) for females. There was also the dose-dependent relationship in the sex-specific analysis (*p* = 0.01 for both males and females).

### 3.5. Sensitivity Analysis

After excluding total cholesterol from the comorbidity risk score, a GFR ≥90 mL/min/1.73 m^2^ was the most substantial contributor to the comorbidity risk score (score = 10), followed by hypertension (score = 8), prediabetes and diabetes (score = 5), elevated blood pressure (score = 2), and GFR 60–89 mL/min/1.73 m^2^ (score = 2) (Table A3). Kaplan–Meier estimates are presented (Figure A2) and subjects who scored 9–10 and 11–23 were at significantly higher risks of incident cancer than those who scored 0–2, with HRs (95% CIs) of 1.64 (1.03, 2.59) and 2.12 (1.27, 3.55), respectively (Table A4). The significant finding was observed in males who scored 3–8, 9–10, or 11–23 despite the large standard errors, but not in females (Table A4).

In the sensitivity analysis of participants regardless of a time to cancer diagnosis, a GFR ≥90 mL/min/1.73 m^2^ was still the primary contributor to the comorbidity risk score (score = 9), followed by hypertension (score = 7), normal total cholesterol (score = 6), prediabetes and diabetes (score = 2), GFR 60–89 mL/min/1.73 m^2^ (score = 2), and elevated blood pressure (score = 1) (Table A5). Kaplan–Meier estimates are presented (Figure A3), and compared to subjects who scored 0–4; those who scored 11–24 were at a significantly higher risk of incident cancer, with HR = 1.81 and 95% CI = 1.23, 2.67 (Table A6). A significant finding was observed in females who scored 11–24 (HR = 1.79, 95% CI = 1.10, 2.93), but not in males (Table A6).

## 4. Discussion

In this prospective cohort study, we found nonsignificant associations between comorbidity markers, including blood pressure, fasting glucose, and the GFR, and cancer incidence. However, after combining the simultaneous effect of the four comorbidities into one risk score, the risk of incident cancer was 115% higher among subjects who scored 16–30 than among those who scored 0–8. Sex-specific subgroup analyses also showed significant associations. The relationship was observed to be dose-dependent.

Although categories of blood pressure (*p* < 0.001), fasting glucose (*p* < 0.001), and chronic kidney disease (*p* = 0.01) were significantly different among total cholesterol groups (Table A7), a further analysis of excluding total cholesterol from the comorbidity risk score was performed. The comorbidity risk score for each category was, therefore, lower than those in the main analysis (Table A4 and Table 5). The small number of cases among males who scored 0–2 might result in the large 95% CIs. Furthermore, nonsignificant associations were observed for the subgroup analysis of females. Thus, considering total cholesterol with other chronic conditions was necessary to detect the simultaneously significant effect of comorbidities on sex-specific cancer incidence.

In the sensitivity analysis of participants regardless of time to cancer diagnosis, compared to findings from the main analysis, the point estimates and their 95% CIs for the association between individual comorbidities and cancer incidence tended to be closer to null (Table 3 and Table A5). The risk point according to the chronic conditions of blood pressure, total cholesterol, and fasting glucose, therefore, tended to increase after excluding subjects with cancer diagnosis within 1 year (Table 3 and Table A5). Additionally, while males whose comorbidity risk scores were in the highest category had a significantly higher risk of cancer (HR = 2.39, 95% CI = 1.18, 4.84, Table 5), the association was not observed when including early cases in the analysis (HR = 1.79, 95% CI = 0.92, 3.52, Table A6). These changes can be partially explained by the reverse causality effect of subjects with undetected tumors at baseline [30,31]. Thus, excluding early cases from our analysis minimized the underestimate of effect associations.

Regarding the dietary score, this study applied the GGM, which has been widely used in research on genetics, psychology, and climate. Iqbal et al. also used the GGM to identify major dietary patterns and investigate their associations with type 2 diabetes, cardiovascular disease, and cancer [32]. However, the quantitative measurement of the dietary score using centrality indices of nodes in the network has still not been performed. Other data-derived methods, such as principal component analysis and reduced rank regression, have been manipulated for data dimension reduction purposes [33]. However, the GGM is considered a novel approach to describe the partial correlations between food groups [32].

Several dietary scores have been developed to elucidate the role of dietary intake in the etiology of cancer [34,35]. Recently, Lassale et al. systematically reviewed the available dietary indices, which were used to investigate the association of diet with depression; these included the Mediterranean diet, the healthy eating index, dietary approaches to stop hypertension, and the dietary inflammatory index [36]. Although different approaches might be associated in the same direction with a health outcome, the posteriori scores of the factor analysis method were reported to be better than the priori scores of common methods in the evaluation of coronary heart disease risk [37].

In terms of the comorbidity risk score, the Charlson comorbidity index and the Elixhauser score are commonly used to determine the severity of health conditions [38,39]. The Charlson comorbidity index was originally developed with 19 conditions, based on general internal medicine service claims and the mortality rate of 607 patients over one month [40]. The Elixhauser comorbidity index consists of 30 conditions based on the International Classification of Diseases diagnosis codes and was developed to predict hospital resource use and in-hospital mortality [41]. However, data on some rare conditions might not be available for our dataset from the general healthy population; thus, we developed the data-driven approach with a risk score algorithm. The comorbidity risk score derived from the Cox model has been widely applied in several studies to predict the risk of hepatocellular carcinoma in the US, Taiwanese, Japanese, Chinese, and Korean populations [42].

Despite its strengths, this study has several limitations. First, we used data from a cancer screening program of the National Cancer Center; thus, this might not represent the whole general Korean population. Second, macro- and micronutrients were not involved in our dietary score assessment models. As nutrients generally did not have the same unit, we were unable to combine all the nutrients in a single dietary score. Because food intake was considered the covariate and we focused on the effects of comorbidities, the daily amount of food consumed was used to reflect the dietary status of individuals. Third, the number of incident cases was limited due to the short follow-up duration, which resulted in larger 95% CIs for the sex-specific subgroups than for the entire study population. Additionally, we were not able to perform subgroup analyses by cancer type. Last, the history of chronic diseases and drugs currently being taken were obtained by subject self-report only and not from medical records. However, the information was obtained via several separate but related questions to minimize recall bias. Regarding medical history, subjects were asked about their names, their duration of taking drugs in years and months, and their frequency of taking drugs. Additionally, subjects were asked about the year when a chronic disease was first diagnosed and whether the disease had been treated or not. These self-reported data were combined with blood test results to classify the comorbidity status.

The evidence for the association between high cholesterol and cancer incidence is controversial. In addition, cholesterol was reported to activate oncogenic Hedgehog signaling and mTORC1, which might result in the differentiation and proliferation of cells, tumor formation, and metastasis [43]. Furthermore, abnormal cholesterol levels could impact the structure of lipid rafts, which is known to be a vital structure involved in cancer signaling [43]. Epidemiological studies also support positive associations between cholesterol levels and breast, prostate, and colorectal cancers [44]. However, low cholesterol was reported to be associated with a higher risk of cancer in Korean, Japanese, Taiwanese, and Caucasian populations [4,23]. In contrast, Kim et al. performed a meta-analysis of approximately 65,000 individuals from randomized controlled trials, and no beneficial effect of cholesterol-lowering medications on cancer prevention was found (pooled relative risk 0.97, 95% CI = 0.92, 1.02) [45]. Subgroup analysis by cancer type, statin type, and country showed nonsignificant associations [45]. Nevertheless, our study showed a significant relationship between total cholesterol and overall cancer incidence.

Meta-analyses reported an approximately 30% increased risk of colorectal cancer among the population with diabetes mellitus [46,47]. The Korean Multicenter Cancer Cohort study conducted from 1993 to 2005 with a median follow-up of 12 years found that the joint consideration of fasting glucose and a history of diabetes mellitus was not significantly associated with colorectal cancer (HR = 1.54, 95% CI = 0.97, 2.43) [48], which was consistent with our findings. However, an increased risk of colorectal cancer was observed in diabetes patients when the duration of the follow-up was more than five years (HR = 1.61, 95% CI = 1.02, 2.56) [48].

Evidence from a recent comprehensive meta-analysis showed significant associations between blood pressure and the risks of kidney, colorectal, and breast cancers, but not the risks of cancer of the stomach, gallbladder, pancreas, lung, cervix, prostate, bladder, and brain [49]. Because of the limited number of cancer cases, we did not perform a subgroup analysis by cancer type. However, we still found a significant association between hypertension and total cancer incidence. Despite the nonsignificant association, the difference between the point estimates of the coefficient for the effect of elevated blood pressure and the coefficient for a one-year increase in age was substantial, and elevated blood pressure still contributed to the risk score.

In the current study, the highest comorbidity risk score was assigned to those who had a normal GFR ≥90 mL/min/1.73 m^2^. Several studies have suggested that a high GFR is associated with an increased risk of cancer [4]. This could be explained by patients whose kidney function was mildly or moderately impaired, as a decrease in the GFR is more likely to develop due to other kidney-related diseases and not cancer. Similarly, although we did not observe a significant association between the GFR and cancer incidence in our study, a high GFR was assigned a high-risk score.

## 5. Conclusions

In summary, the current study found that comorbidities had a joint dose-dependent impact on cancer incidence. These findings may be helpful for the development of cancer prevention programs targeting the management of comorbidities. Further population-based prospective studies with substantial follow-up periods are needed to confirm the association among different cancer subtypes.

## Figures and Tables

**Figure 1 cancers-12-01834-f001:**
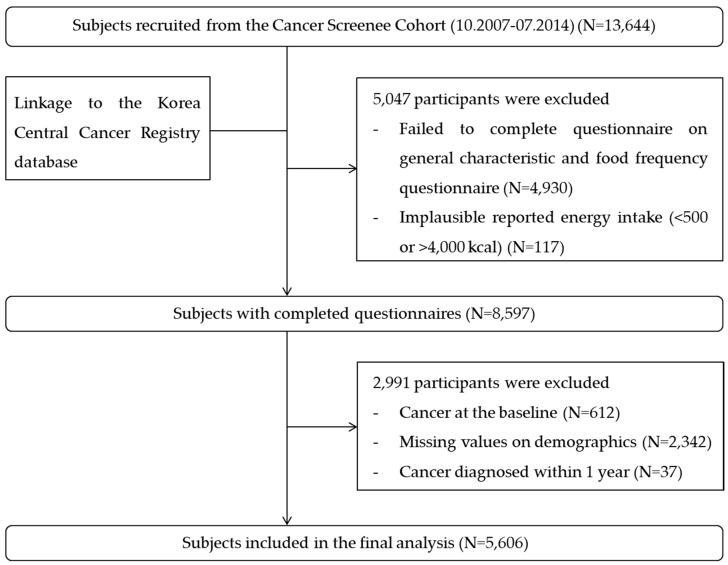
Flowchart of the subject recruitment process.

**Figure 2 cancers-12-01834-f002:**
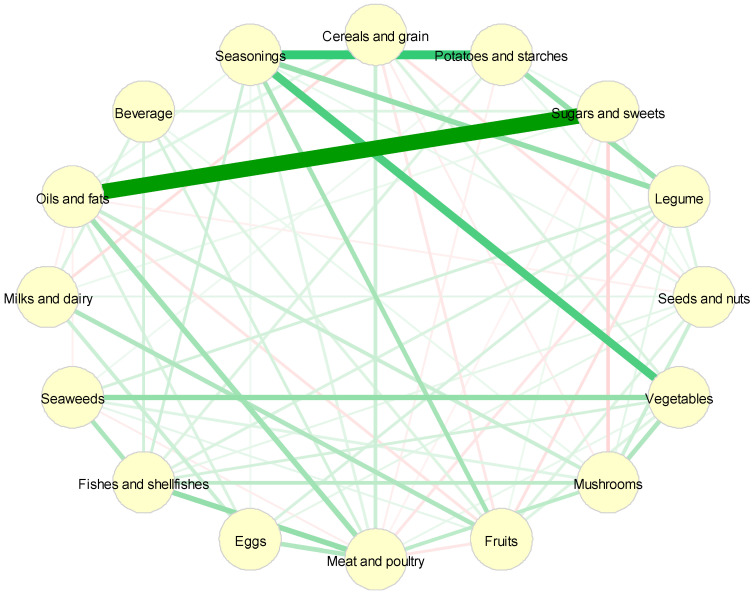
Network of dietary intake derived by Gaussian graphical models.

**Figure 3 cancers-12-01834-f003:**
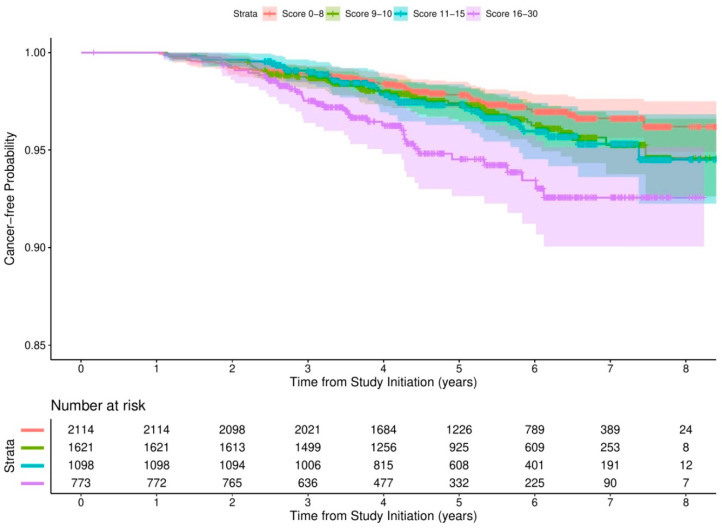
Kaplan–Meier estimates of cancer-free probability by comorbidity risk scores.

**Table 1 cancers-12-01834-t001:** Adjacency matrix for partial correlation of dietary intake of 16 food groups.

	Cereals and Grains	Potatoes and Starches	Sugars and Sweets	Legumes	Seeds and Nuts	Vegetables	Mushrooms	Fruits	Meat and Poultry	Eggs	Fish and Shellfish	Seaweed	Milk and Dairy	Oils and Fats	Beverages	Seasonings
**Cereals and grains**																
**Potatoes and starches**																
**Sugars and sweets**		0.05														
**Legumes**		0.22														
**Seeds and nuts**	−0.08	0.05	0.05	0.08												
**Vegetables**	0.08															
**Mushrooms**	−0.05		−0.12	0.07	0.12	0.18										
**Fruits**	−0.07		0.04	−0.08	0.08	0.08										
**Meat and poultry**	0.11	−0.05		−0.04		−0.07	0.05	0.15								
**Eggs**				−0.07	0.09			0.06	0.17							
**Fish and shellfish**		0.08		0.09	0.06	0.10	0.15		0.23							
**Seaweed**		0.04		0.10		0.23	0.07	0.06	−0.06		0.19					
**Milk and dairy**	−0.09		0.05		0.07			0.18		0.12						
**Oils and fats**	0.07		0.70		−0.05		0.12	−0.08	0.20	0.08		−0.05	−0.04			
**Beverages**			0.08				0.05	0.08	0.10		0.10		0.08			
**Seasonings**		0.37		0.23	0.06	0.34		0.19	0.07	0.04	0.11			0.05		

**Table 2 cancers-12-01834-t002:** Dietary intake of study population and eigenvector centrality of 16 food groups.

Food Group	Daily Intake (g/day)				Eigenvector Centrality
	Total (N = 5,606)	Cancer (N = 176)	Noncancer (N = 5430)	*p*-Value	
Cereals and grains	582.77 ± 213.71	564.0 ± 188.5	583.4 ± 214.5	0.18	0.09
Potatoes and starches	44.49 ± 43.74	43.1 ± 45.9	44.5 ± 43.7	0.68	0.68
Sugars and sweets	5.02 ± 5.24	4.6 ± 5.2	5.0 ± 5.2	0.27	0.51
Legumes	59.84 ± 67.42	57.4 ± 54.2	59.9 ± 67.8	0.55	0.59
Seeds and nuts	5.52 ± 9.98	4.8 ± 12.7	5.5 ± 9.9	0.42	0.32
Vegetables	312.12 ± 204.07	322.4 ± 209.2	311.8 ± 203.9	0.50	0.76
Mushrooms	9.05 ± 13.48	10.3 ± 16.4	9.0 ± 13.4	0.29	0.50
Fruits	220.65 ± 255.47	225.4 ± 246.1	220.5 ± 255.8	0.80	0.27
Meat and poultry	58.62 ± 51.63	55.8 ± 50.3	58.7 ± 51.7	0.46	0.48
Eggs	17.33 ± 18.04	17.0 ± 18.3	17.3 ± 18.0	0.82	0.30
Fish and shellfish	39.99 ± 34.98	40.0 ± 35.2	40.0 ± 35.0	>0.99	0.70
Seaweed	2.25 ± 2.37	2.4 ± 2.4	2.2 ± 2.4	0.53	0.45
Milk and dairy	108.63 ± 137.91	101.7 ± 139.0	108.9 ± 137.9	0.50	0.13
Oils and fats	3.83 ± 3.81	3.5 ± 3.6	3.8 ± 3.8	0.19	0.61
Beverages	75.15 ± 110.42	86.2 ± 126.4	74.8 ± 109.9	0.24	0.25
Seasonings	17.42 ± 14.80	17.2 ± 13.7	17.4 ± 14.8	0.81	1.00

Data for daily intake are presented as mean (standard deviation). *p*-values were obtained by performing a *t*-test.

**Table 3 cancers-12-01834-t003:** Comorbidity biomarkers and the risk of cancer incidence.

Comorbidity Biomarkers	Population (N = 5606)	Incidence (N = 176)	Adjusted HR (95% Confidence Interval (CI)) ^a^	Fully Adjusted HR (95% CI) ^b^	Risk Point ^c^
**Blood pressure (mmHg)**					
Normal	1662	38	1 (ref)	1 (ref)	0
Elevated	2225	59	1.07 (0.71, 1.61)	1.12 (0.74, 1.70)	2
Hypertension	1719	79	1.53 (1.02, 2.29)	1.56 (1.02, 2.39)	8
**Total cholesterol (mg/dL)**					
Low	1492	47	1.23 (0.86, 1.77)	1.31 (0.91, 1.88)	5
Normal	1141	42	1.39 (0.96, 2.01)	1.51 (1.03, 2.19)	8
Elevated	2793	87	1 (ref)	1 (ref)	0
**Fasting glucose (mmol/L)**					
Normal	5029	147	1 (ref)	1 (ref)	0
Prediabetes and diabetes	577	29	1.40 (0.93, 2.09)	1.32 (0.87, 2.00)	5
**Glomerular filtration rate (mL/min/1.73 m^2^)**					
<60	268	13	1 (ref)	1 (ref)	0
60–89	4459	142	0.99 (0.55, 1.78)	1.11 (0.62, 2.02)	2
≥90	739	21	1.41 (0.69, 2.89)	1.65 (0.80, 3.43)	9

**^a^** Adjusted for age; **^b^** Adjusted for age, sex, marital status, education, employment, monthly income, smoking, drinking, physical activity, body mass index, dietary score, and other chronic diseases; **^c^** Calculated by dividing the coefficient of each level of comorbidity biomarkers by the coefficient of age in the fully adjusted model.

**Table 4 cancers-12-01834-t004:** Baseline characteristics of the study population.

	Cancer Status			Comorbidity Risk Score				
	Cancer (N = 176)	Noncancer (N = 5430)	*p*-Value	Score 0–8 (N = 2114)	Score 9–10 (N = 1621)	Score 11–15 (N = 1098)	Score 16–30 (N = 773)	*p*-Value
**Age (years)**	55.8 (8.7)	52.5 (8.2)	<0.001	51.6 (8.1)	52.8 (8.3)	53.3 (8.6)	53.7 (8.0)	<0.001
<50	44 (25.0%)	2051 (37.8%)	<0.001	880 (41.6)	589 (36.3)	395 (36.0)	231 (29.9)	<0.001
50–54	41 (23.3%)	1272 (23.4%)		518 (24.5)	377 (23.3)	229 (20.9)	189 (24.5)	
55–59	23 (13.1%)	933 (17.2%)		332 (15.7)	291 (18.0)	182 (16.6)	151 (19.5)	
≥60	68 (38.6%)	1174 (21.6%)		384 (18.2)	364 (22.5)	292 (26.6)	202 (26.1)	
**Sex**								
Female	105 (59.7%)	3420 (63.0%)	0.41	1553 (73.5)	1030 (63.5)	579 (52.7)	363 (47.0)	<0.001
Male	71 (11.9%)	2010 (37.0%)		561 (26.5)	591 (36.5)	519 (47.3)	410 (53.0)	
**Marital status**								
Married, cohabitant	155 (88.1%)	4675 (86.1%)	0.53	1822 (86.2)	1386 (85.5)	953 (86.8)	669 (86.5)	0.79
Others	21 (11.9%)	755 (13.9%)		292 (13.8)	235 (14.5)	145 (13.2)	104 (13.5)	
**Education**								
<High school	27 (15.3%)	718 (13.2%)	0.37	239 (11.3)	218 (13.4)	141 (12.8)	147 (19.0)	<0.001
High school graduate	58 (33.0%)	2059 (37.9%)		796 (37.7)	626 (38.6)	416 (37.9)	279 (36.1)	
≥ College	91 (51.7%)	2653 (48.9%)		1079 (51.0)	777 (47.9)	541 (49.3)	347 (44.9)	
**Employment**								
Employed	153 (86.9%)	4994 (92.0%)	0.02	1980 (93.7)	1503 (92.7)	973 (88.6)	691 (89.4)	<0.001
Unemployed	23 (13.1%)	436 (8.0%)		134 (6.3)	118 (7.3)	125 (11.4)	82 (10.6)	
**Monthly income (**South Korean won (**KRW))**								
<2 million	45 (25.6%)	1182 (21.8%)	0.48	426 (20.2)	352 (21.7)	264 (24.0)	185 (23.9)	0.13
2–4 million	69 (39.2%)	2207 (40.6%)		860 (40.7)	668 (41.2)	436 (39.7)	312 (40.4)	
≥4 million	62 (35.2%)	2041 (37.6%)		828 (39.2)	601 (37.1)	398 (36.2)	276 (35.7)	
**Smoking**								
Never	115 (65.3%)	3600 (66.3%)	0.97	1571 (74.3)	1085 (66.9)	635 (57.8)	424 (54.9)	<0.001
Past	39 (22.2%)	1166 (21.5%)						
Current	22 (12.5%)	664 (12.2%)		321 (15.2)	346 (21.3)	313 (28.5)	225 (29.1)	
**Drinking**								
Never	68 (38.6%)	2117 (39.0%)	0.80	909 (43.0)	634 (39.1)	392 (35.7)	250 (32.3)	<0.001
Past	14 (8.0%)	362 (6.7%)		131 (6.2)	113 (7.0)	78 (7.1)	54 (7.0)	
Current	94 (53.4%)	2951 (54.3%)		1074 (50.8)	874 (53.9)	628 (57.2)	469 (60.7)	
**Physical activity**								
No	71 (40.3%)	2405 (44.3%)	0.34	936 (44.3)	726 (44.8)	472 (43.0)	342 (44.2)	0.83
Yes	105 (59.7%)	3025 (55.7%)		1178 (55.7)	895 (55.2)	626 (57.0)	431 (55.8)	
**Body mass index (kg/m^2^)**								
<23	69 (39.2%)	2448 (45.1%)	0.14	1131 (53.5)	707 (43.6)	404 (36.8)	275 (35.6)	<0.001
23–24.9	46 (26.1%)	1455 (26.8%)		526 (24.9)	431 (26.6)	328 (29.9)	216 (27.9)	
≥25	61 (34.7%)	1527 (28.1%)		457 (21.6)	483 (29.8)	366 (33.3)	282 (36.5)	
**Dietary score** **(g/day)**	542.8 (285.3)	537.4 (279.1)	0.80	532.8 (275.9)	536.0 (276.1)	551.3 (301.3)	534.5 (261.5)	0.33
Light eating	58 (33.0%)	1811 (33.3%)	0.53	741 (35.1)	529 (32.6)	352 (32.1)	247 (32.0)	0.25
Normal eating	65 (36.9%)	1803 (33.2%)		704 (33.3)	537 (33.1)	356 (32.4)	271 (35.1)	
Heavy eating	53 (30.1%)	1816 (33.5%)		669 (31.6)	555 (34.2)	390 (35.5)	255 (33.0)	

Data are presented as counts (percentages) for categorical variables and means (standard deviations) for continuous variables. *p*-values were obtained by performing chi-square tests for categorical variables and *t*-tests or an ANOVA for continuous variables.

**Table 5 cancers-12-01834-t005:** Comorbidity scores and the risk of cancer incidence.

	Population (N = 5606)	Incidence (N = 176)	Crude HR (95% CI)	Adjusted HR (95% CI) ^a^	Fully Adjusted HR (95% CI) ^b^
**All subjects**					
Score 0–8	2114	53	1 (ref)	1 (ref)	1 (ref)
Score 9–10	1621	51	1.28 (0.87, 1.88)	1.22 (0.83, 1.79)	1.23 (0.84, 1.82)
Score 11–15	1098	35	1.32 (0.86, 2.02)	1.23 (0.79, 1.90)	1.24 (0.80, 1.93)
Score 16–30	773	37	2.27 (1.49, 3.46)	2.12 (1.37, 3.27)	2.15 (1.39, 3.31)
*P*-trend			<0.001	<0.001	<0.001
**Males ^c^**					
Score 0–8	561	13	1 (ref)	1 (ref)	1 (ref)
Score 9–10	591	20	1.48 (0.74, 2.98)	1.30 (0.64, 2.64)	1.32 (0.65, 2.68)
Score 11–15	519	17	1.47 (0.71, 3.02)	1.24 (0.59, 2.57)	1.25 (0.60, 2.60)
Score 16–30	410	21	2.67 (1.34, 5.34)	2.32 (1.15, 4.70)	2.39 (1.18, 4.84)
*P*-trend			0.001	0.01	0.01
**Females ^c^**					
Score 0–8	1553	40	1 (ref)	1 (ref)	1 (ref)
Score 9–10	1030	31	1.20 (0.75, 1.92)	1.17 (0.73, 1.87)	1.17 (0.73, 1.88)
Score 11–15	579	18	1.26 (0.72, 2.21)	1.20 (0.69, 2.10)	1.20 (0.69, 2.11)
Score 16–30	363	16	2.04 (1.14, 3.65)	1.99 (1.11, 3.58)	1.99 (1.11, 3.59)
*P*-trend			0.01	0.01	0.01

**^a^** Adjusted for age, sex, education, employment, income, smoking, drinking, and body mass index; **^b^** Additionally adjusted for marital status, physical activity, and eating behavior; **^c^** Not adjusted for sex in the adjusted and fully adjusted models.

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
