# Peer review of "Comorbidity Risk Score in Association with Cancer Incidence: Results from a Cancer Screenee Cohort"

_cancers, 2020, doi:10.3390/cancers12071834_

Round 1

Reviewer 1 Report

I think that the authors addressed my comments.

Reviewer 2 Report

Seems the authors adressed all the comments raised. at this point i donot have any more comments.

This manuscript is a resubmission of an earlier submission. The following is a list of the peer review reports and author responses from that submission.

Round 1

Reviewer 1 Report

This is an important and understudied topic. The introduction is very clear.

Methods:

  1. Diet intake categorization is not clear. Low, medium, and high seem to be related to the node weights of food groups. What is the physiological meaning of such categorization? For example, high energy intake vs. low energy intake has a physiological meaning. It is not clear what physiology stands behind this categorization.
  2. Gaussian graphical model: this part of the work should be complemented by figures showing progressive selection.
  3. The following also needs more explanation: “Then, the comorbidity scores were obtained by dividing the above-calculated regression coefficients for comorbidity with the coefficient for a one-year increase in age.”
  4. The reference level for cholesterol is not justified. In general, the previous studies did not support the association between cholesterol and cancer risk (Annu Rev Nutr. 1992;12:391-416).
  5. The following additional analyses should be presented:
  6. Exclude cholesterol form the risk score and show the results.
  7. Exclude cancer cases that have been diagnosed during the first year from the baseline and present the results. Cancer has a long latency, which means that the tumor was already present but clinically undetected at baseline for the cases which are identified during year 1. Since the risk score is based on biomarkers, it is important to show that undetected tumor did not influence the risk score, which would constitute reverse causality.

Discussion:

  1. Discussion still does not explain what the physiological meaning of the scoring is.
  2. Discussion on cholesterol mentions cholesterol-lowering drugs. However, the drugs themselves but not their effect on cholesterol. Taking this into account, if the authors have information on cholesterol-lowering drugs, this should be used in the analysis. However, it is just better and a more clean approach to using the lower category as a reference and then exclude cholesterol from the risk score and present the results.

Author Response

Thank you for the critical comments that are very helpful for our manuscript.

We have revised the manuscript according to your suggestion and recommendation.

Please see the attachment for more details.

Reviewer 2 Report

The rationale and idea of querying the comorbidity risk score for potential cancer incidence or development is interesting.  However, at present, the submitted manuscript has multiple falws, which need to be addressed before it is accepted for publication.

  • The comorbidity risk score was associated with cancer incidence in a dose-dependent manner (HR=1.74, 95% CI=1.15, 2.63 for those 19 scoring 12-20 vs. those scoring 0-3, P-trend=0.003). Subgroup analysis tended to show borderline relationships (HR=2.02, 95% CI=0.99, 4.10 for males, and HR=1.71, 95% CI=1.00, 2.92 for females) but still in a dose-dependent manner (P-trend<0.05). There was a joint impact of comorbidities on cancer incidence in a dose-dependent manner. If we understood right, most of the risk is because of fat and sugar intake. There a lot of epidemiological study which states that high lipid content in their food is associated with cancer risk. In this scenario, the present manuscript did not add much to the existing literature.
  • Further, as the difference is small though significant, it need to be consider multiple cofactors
  • In general, cancer is an age-related disease. It is surprising to see there is no mention about that.
  • Second one – the follow up is ~ 5 years – whereas the tumor development is decade long process if it is due to environment risk related. How the manuscript is going to address this disparity. The study should consider longer follow-up.
  • In many countries, over the counter medicine is not well organized and many times, it is not coming into the picture of record. The study did not consider the over the counter medicine influence in their study population.
  • Whether the study adjusted the study population to preexisting diabetes or obese? And BMI taken into account.  
  •  As mentioned by the author, the macro- and micronutrients were not taken into account, which is a critical component.
  • Did the study consider any income based association? In general, the people with higher wages are better of in their nutrition’s value. Or did the authors adjusted.
  • Any association with alcohol intake and smoking?

Author Response

(The authors gave the same response as above.)

Reviewer 3 Report

In this manuscript, Hoang et al. investigate the impact of comorbidities on cancer incidence by applying a gaussian model to calculate a dietary score for patients. In order to generate a comorbidity score they used four comorbidity markers: blood pressure, total cholesterol, fasting glucose, and the glomerular filtration rate (GFR). The authors report that while individual comorbidities were not found to significantly impact cancer incidence, multiple comorbidities, as observed in 883 out of 5643 patients, was significantly associated with cancer. Thus, monitoring for these comorbidities in patients can help predict cancer incidence, such that early interventions can be implemented to stall tumorigenesis.

The manuscript is generally well written, devoid, for the most part of grammatical errors. The methods are well described and results well laid out, except a few areas where their inclusion of certain subjects is not explained thoroughly (please see points 1 and 2 under Methods and Results section of the review). A recurring theme across the manuscript is a lack of discussion of implications of their findings. I have detailed several such instances below where I find a lack of a succinct and implicit description of what their observations and findings mean.

Abstract:

What are the major findings and significance of these findings in this study? The authors need to highlight this in one or two sentences at the end of the abstract.

Introduction:

  1. Lines 31-32 suggest that certain comorbities evolve concomitantly (and independently?) with cancer. However, lines 37-42 suggest that conditions such as diabetes, hypertension etc can increase the risk of cancer in these patients. It is confusing whether the authors are trying to address predisposition to cancer or co-evolution of cancer with these other diseases?
  2. Line 52: what are “concomitant comorbidities”? aren’t comorbidities by definition concomitant?
  3. Lines 44-50: Tu et al conducted a study where they looked at the impact of fruit and vegetable intake on cancer incidence. In other words, the impact of dietary behavior on cancer incidence. The authors then state in lines 51-53 that dietary behavior and lifestyles are regionally varied. However, the final statement completely disconnects from either dietary or lifestyle behaviors. Are the authors trying to distinguish this study from Tu et al by stating that they are looking at dietary behaviors in a different region? Tu et al also studied the impact of comorbities on cancer incidence. So I am not sure what the authors are doing differently in this study. The authors need to state that clearly in these last few sentences.

Methods and Results:

  1. Line 60: “A total of 8,597 individuals were identified at baseline. Including those with self-reported cancer or those previously diagnosed with cancer” – what does self-reported mean? That these patients were not clinically diagnosed with cancer? If so, how was their medical history reliably assessed?
  2. Line 71: “blood pressure, total cholesterol, fasting glucose, and the glomerular filtration rate (GFR), together with self-reported hypertension and diabetes.” – self reported again implies clinically undiagnosed conditions. How were these reliably factored into the risk assessment?
  3. Lines 106-112: the authors state their observations here but they also need to state the implications of these observations. What does it mean when “The strongest regularized partial correlation was observed between ‘oils and fats’ and ‘sugars and sweets’”. Does this mean that oils and fats are most often consumed with sugars sweets? And do these food groups represent the highest risk of predisposition to cancer? What do the scores of 0.70, 0.37 etc indicate? The implication of this observation needs to be stated implicitly.
  4. Lines 130-131: “Cancer subjects were observed to be significantly older than noncancer participants, with age at baseline in the two groups of 55.6±8.5 years and 52.5±8.2, respectively (p<0.001)” – while this is statistically significant, is the age of 55.6 clinically and physiologically that different from 52.5? Since comorbidities are being discussed, are patients of age 55.6 more likely to have higher total cholesterol, blood glucose and GFR compared to patients of age 52.5?
  5. Table 4: I am confused about the data presentation here. For example, for the GFR category, out of 4629 patients with GFR values of 60-89, 172 had incidence of cancer. Thus 3.7% of the population demonstrated cancer incidence. Whereas, out of 744 patients with GFR of more than 90, 25 patients showed incidence of cancer. Thus 3.36% of the population demonstrated cancer incidence. Doesn’t that indicate that a higher GFR does not greatly increase predisposition to cancer? Please clarify. If so, please make this clear in the results section for all comorbidities.

Author Response

(The authors gave the same response as above.)

Round 2

Reviewer 1 Report

The following needs further clarifications:

  1. It is still not clear what are the dietary groups. The authors said it is high consumption in g/day of GGM-derived dietary intake. What does this mean - what kind of food is GGM-derived intake? This is an abstract construct without any physiological meaning.
  2. Sensitivity analysis with exclusion of cholesterol has been conducted. However, in the presentation of the results, the authors did not compare the findings of the models before and after the exclusion of cholesterol. Such a comparison is the reason why sensitivity analysis is done.
  3. Sensitivity analysis regarding the time of cancer diagnosis was not done correctly. It is important to exclude "early cases" - the cases diagnosed during the first year after inclusion in the study.

Reviewer 2 Report

I sid not see the other confounding factors considered in the attached manuscript.

if the authors provide the additional 16 questionaries (any over the counter medication, any preivious diagnosis etc), which  the authors considered.  in the supplementary or in the main text as a table, it would help the researcher in the future to design such a study overall to improve the health of people.

The manuscript can be accepted by the condition of providing the complete details of other co-factors considered.

Reviewer 3 Report

The authors have made the requisite changes and the manuscript has improved significantly.